# Probabilities of Chat LLMs Are Miscalibrated but Still Predict Correctness on Multiple-Choice Q&A

## Abstract

We study 14 large language models (LLMs) fine-tuned for chat and find that their maximum softmax probabilities (MSPs) are consistently miscalibrated on multiple-choice Q&A. However, those MSPs might still encode useful uncertainty information. Specifically, we hypothesized that wrong answers would be associated with smaller MSPs compared to correct answers. Via rigorous statistical testing, we show that this hypothesis holds for models which perform well on the underlying Q&A task. We also find a strong direction correlation between Q&A accuracy and MSP correctness prediction, while finding no correlation between Q&A accuracy and calibration error. This suggests that within the current fine-tuning paradigm, we can expect correctness prediction but not calibration to improve as LLM capabilities progress. To demonstrate the utility of correctness prediction, we show that when models have the option to abstain, performance can be improved by selectively abstaining based on the MSP of the initial model response, using only a small amount of labeled data to choose the MSP threshold.

## 1 Introduction

Large language models (LLMs) have demonstrated profound capabilities in many domains, but continue to sometimes generate plausible-sounding false responses (Huang et al., 2023). In one high-profile case, an LLM-based system invented a litany of nonexistent court cases, leading to formal sanctions for two lawyers (Mangan, 2023). Although ongoing work has reduced the rate of these mistakes,[1] LLMs will inevitably face situations that surpass the boundaries of their existing knowledge. In those situations, it is unrealistic to expect these models (or any intelligent agents, including humans) to always make perfect decisions. Rather than confidently misleading users, LLMs should be able to detect unfamiliar situations and act cautiously (e.g., decline to answer).

In this paper, we study whether LLMs can determine the correctness of their own answers to multiple-choice questions. If so, this would directly enable LLMs to decline to answer when they are likely to be incorrect. Prior work fine-tunes LLMs with labeled data to recognize questions beyond their knowledge (Kadavath et al., 2022; Zhang et al., 2023). However, this approach is costly and could interfere with other capabilities of the model. More broadly, such approaches do not answer the fundamental scientific question of whether LLMs already possess the necessary information to decide when to abstain. In other words, can this ability be *invoked* rather than *learned*?

We investigate the maximum softmax probability (MSP) as a potential source of innate uncertainty information. Specifically, our goal is to understand the relationship between the MSP of an LLM response and the correctness of that response.

We evaluate 14 LLMs fine-tuned for chat (henceforth "chat LLMs" for brevity) on five different Q&A datasets with two different phrasings (Figure 1). These LLMs cover a range of sizes, capabilities and architectures, and include both open-source and proprietary models. To our knowledge, our work is the most comprehensive study on LLM correctness-awareness. This comprehensiveness bolsters the robustness of our claims, but also enables *cross-model* comparisons which reveal novel insights.

---

[1] See, for example, `https://huggingface.co/spaces/hallucinations-leaderboard/leaderboard`.

Below is a multiple-choice question. Choose the letter which best answers the question. Keep your response as brief as possible; just state the letter corresponding to your answer, followed by a period, with no explanation.

Question:
In the nitrogen cycle, nitrogen can return to the lithosphere directly from the atmosphere by
A. lightning.
B. cellular respiration.
C. air pollution.
D. condensation.

Response:

You will be asked a multiple-choice question. Respond with the letter which corresponds to the correct answer, followed by a period. There is no need to provide an explanation, so your response should be very short. Now here is the question:

In the nitrogen cycle, nitrogen can return to the lithosphere directly from the atmosphere by
A. lightning.
B. cellular respiration.
C. air pollution.
D. condensation.

Answer:

Figure 1: The two prompt phrasings we used, demonstrated on an example question.

## 1.1 CALIBRATION OF THE MAXIMUM SOFTMAX PROBABILITY

We first ask whether the MSP is *calibrated* (DeGroot & Fienberg, 1983; Nguyen & O'Connor, 2015), meaning that among responses with an MSP of $p\%$, $p\%$ are correct. Calibrated MSPs enable fully unsupervised abstention policies with theoretical guarantees: a calibrated model which answers only when the MSP is higher than $1 - \varepsilon$ guarantees that the chance of an incorrect answer is at most $\varepsilon$. However, we show that this approach is not generally viable for chat LLMs. Figure 2 (left) shows that most LLMs in our study have poorly calibrated MSPs. In particular, the MSPs are consistently overconfident. Furthermore, improved performance on the underlying Q&A task does not necessarily translate to better calibration: Figure 2 (right) shows no correlation between Q&A accuracy and calibration error. This is one of the cross-model comparisons mentioned above.

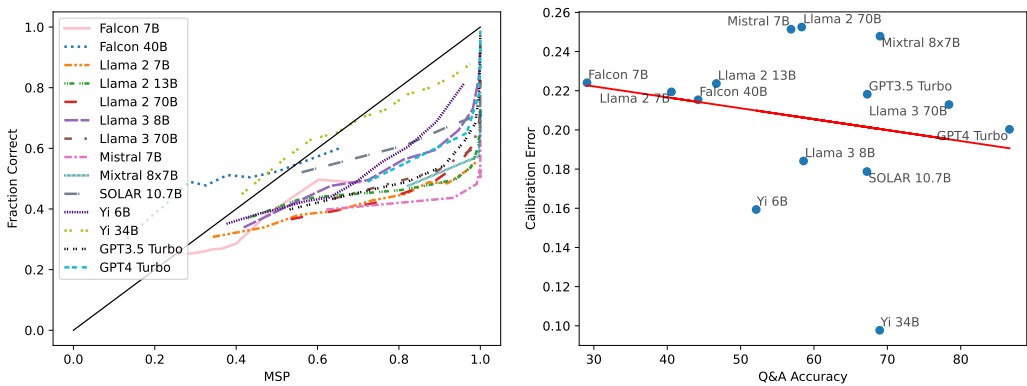

Figure 2: **Left**: calibration graph across all datasets using 10 quantile bins (i.e., each bin contains the same number of data points, and there are 10 data points per model). Most models exhibit clear overconfidence. **Right**: we aggregate the data from the left graph to a single data point per model, capturing the model's expected calibration error (the mean over bins of the absolute difference between the average MSP and the fraction of correct answers in that bin) and average Q&A accuracy. The Q&A accuracy for each model is averaged across the five datasets and two phrasings. The Pearson correlation coefficient is $r = -0.21$ ($p = 0.47$), indicating no correlation.

## 1.2 CORRECTNESS PREDICTION WITHOUT CALIBRATION

Even if the MSP cannot be directly interpreted as the probability of correctness, it might still be predictive of correctness. As a simplified example, consider a model whose MSP is consistently 0.9 for correct responses and 0.8 for incorrect responses. This model is clearly miscalibrated, but its MSP perfectly predicts correctness.

Through rigorous statistical testing, we demonstrate that the MSPs of chat LLMs can indeed predict correctness. Moreover, the predictiveness is stronger for models which perform better on the underlying Q&A task. This second cross-model comparison suggests that the ability to predict correctness will strengthen as the general capabilities of LLMs improve (e.g., by scaling up data and model sizes). The same is not true of calibration, as discussed above. These contrasting results reveal a novel dichotomy between two approaches to uncertainty quantification.

## 1.3 EXPERIMENT DESIGN IN BRIEF

Before detailing our results on correctness prediction, we outline our experiment design. We hypothesized that wrong answers on multiple-choice Q&A tasks would be associated with lower MSPs. We chose multiple-choice Q&A because there is exactly one correct answer: this allows us to study the core hypothesis of whether MSPs are predictive of correctness without having to deal with more complex issues such as degrees of correctness or multiple valid phrasings of the same correct answer.

We evaluated our hypothesis on 14 popular LLMs. Each model was tested on each of the five multiple-choice Q&A datasets used in the Hugging Face Open LLM leaderboard (Beeching et al., 2023), which is the state-of-the-art for benchmarking open-source LLMs. To test our hypothesis in the simplest setting possible, we used a plain zero-shot prompting style, with two different prompt phrasings (Figure 1).

Formally, for each LLM-dataset-phrasing combination, we studied a binary classification task: given a multiple-choice question and the LLM's response, predict whether the response is correct. We hypothesized that the MSP classifier would be able to discriminate between correct and incorrect answers better than random chance. We also studied a classifier based on the maximum pre-softmax logit ("Max Logit").[2]

Our primary success metric was the Area Under the Receiver Operating Characteristic curve (AUROC) (Bradley, 1997). The AUROC of a binary classifier ranges from 0% to 100%, where 0% corresponds to getting every prediction wrong, 50% is random chance, and 100% is perfect classification. AUROC is also equivalent to the probability that a randomly chosen positive instance is ranked higher than a randomly chosen negative instance. Conveniently, AUROC is threshold-independent: it captures the model's performance across the entire range of possible thresholds.[3]

We have attached the code for all of our experiments and analysis as supplementary material.

## 1.4 RESULTS

We computed the AUROC for each possible combination of LLM, dataset, prompt phrasing, and classifier (MSP or Max Logit). We could not compute Max Logit AUROCs for the OpenAI models (GPT3.5 Turbo and GPT4 Turbo) because the OpenAI API only provides softmax probabilities and not pre-softmax logits. This resulted in 260 AUROC data points: 140 for MSP and 120 for Max Logit. Among the 10 LLMs with the best average Q&A accuracy (percentage of questions answered correctly), the AUROC outperformed random chance with $p < 10^{-4}$ for 99/100 MSP data points and 78/80 Max Logit data points (Table 1). The other four LLMs exhibited statistically significant AUROC only for 20/40 MSP data points and 15/40 Max Logit data points. The most accurate LLM – GPT4 Turbo – obtained an impressive average AUROC of 86%. See Section 4 for details.

We also found a strong direct correlation between average Q&A accuracy and average AUROC: the Pearson correlation coefficients for MSP AUROC and Max Logit AUROC were $r = 0.83$ ($p =< 10^{-3}$) and $r = 0.91$ ($p < 10^{-4}$), respectively (Figure 3). This contrasts with calibration error (Figure 2), which exhibited no correlation with Q&A accuracy ($r = -0.21, p = 0.47$).

In addition to demonstrating the predictive power of the MSP and maximum logit, we provide a proof-of-concept for how this information can be leveraged to reduce LLM harm in practice. We analyzed a variant of the original Q&A task where models can also abstain and receive 1 point per correct answer, 0 points per abstention, and $-c$ points per wrong answer. Unlike AUROC, this metric is not threshold independent, so we must choose a specific threshold. To do so, we used 20

---

[2]Max Logit has no notion of calibration, since it is not a probability.

[3]Note that calibration error is also threshold independent.

Table 1: Main AUROC results. AUROC and Q&A values are percentages, averaged over ten data points (five datasets and two phrasings). The $p < 10^{-4}$ columns indicate how many of those ten data points yielded $p$ values below $10^{-4}$ for the null hypothesis that AUROC = 50%. The $p$-values are from the Mann-Whitney $U$ test; see Section 4 for details.

| LLM | Q&A Accuracy | MSP | | Max Logit | |
|-----|-----|-----|-----|-----|-----|
| | | AUROC | $p < 10^{-4}$ | AUROC | $p < 10^{-4}$ |
| Falcon 7B | 29.1 | 51.8 | 1/10 | 51.7 | 0/10 |
| Falcon 40B | 44.2 | 59.6 | 7/10 | 55.1 | 4/10 |
| Llama 2 7B | 40.6 | 57.3 | 6/10 | 55.9 | 5/10 |
| Llama 2 13B | 46.7 | 60.1 | 6/10 | 58.0 | 6/10 |
| Llama 2 70B | 58.3 | 69.4 | 10/10 | 63.3 | 8/10 |
| Llama 3 8B | 58.6 | 71.2 | 10/10 | 68.9 | 10/10 |
| Llama 3 70B | 78.4 | 81.7 | 10/10 | 72.6 | 10/10 |
| Mistral 7B | 56.9 | 63.5 | 10/10 | 62.7 | 10/10 |
| Mixtral 8x7B | 69.0 | 61.6 | 10/10 | 62.4 | 10/10 |
| SOLAR 10.7B | 67.2 | 59.9 | 9/10 | 65.2 | 10/10 |
| Yi 6B | 52.1 | 66.8 | 10/10 | 61.7 | 10/10 |
| Yi 34B | 69.0 | 67.5 | 10/10 | 66.4 | 10/10 |
| GPT3.5 Turbo | 67.3 | 75.7 | 10/10 | – | – |
| GPT4 Turbo | 86.6 | 85.5 | 10/10 | – | – |

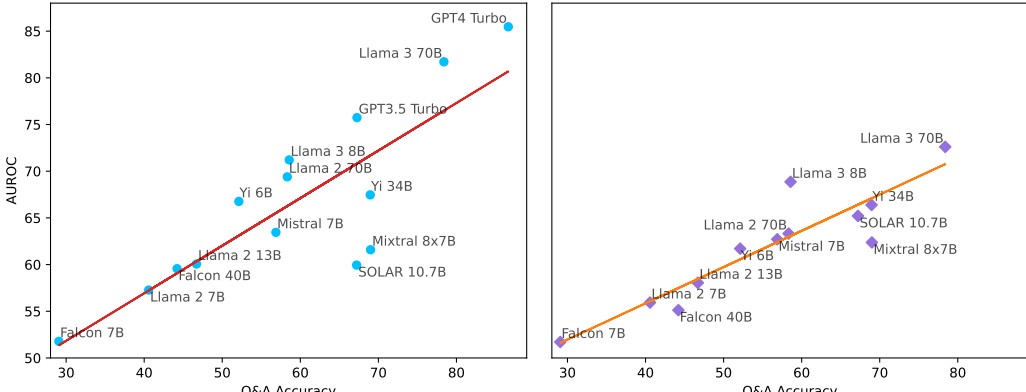

Figure 3: Average AUROC vs average Q&A accuracy for MSP (left) and Max Logit (right). These plots use the same data as Table 1. The Pearson correlation coefficients were $r = 0.83$ ($p < 10^{-3}$) for MSP and $r = 0.91$ ($p < 10^{-4}$) for Max Logit, indicating strong correlations.

randomly chosen labeled data points (i.e., 20 questions and their answers). We found that for both $c = 1$ and $c = 2$, selectively abstaining based on the MSP and/or maximum logit led to substantial improvements over the base models. See Section 5 for details.

In summary, we show the following:

1. LLM MSPs are miscalibrated, and this does not improve as model capabilities improve.

2. LLM MSPs (and maximum logits) still predict correctness, and this *does* improve as model capabilities improve.

3. A small amount of labeled data is sufficient to translate correctness prediction into a simple yet effective abstention method.

Combining with the results of Kadavath et al. (2022), we conclude that fine-tuning degrades calibration of LLMs and this effect is not mitigated as models become more capable. However, this procedure only distorts rather erases uncertainty information in LLMs.

## 2 RELATED WORK

**Kadavath et al. (2022).** The most relevant prior paper is Kadavath et al. (2022), who study LLM correctness-awareness in a variety of domains. There are three key differences between their work and ours.

The first is that they primarily study raw pretrained (i.e., not fine-tuned) LLMs. In particular, their well-known finding that LLMs are well-calibrated applies only to raw pretrained models. In this way, our work complements theirs by showing that their finding of good calibration fails to generalize to LLMs fine-tuned for chat. (They actually briefly study one fine-tuned model and find that it is quite miscalibrated.) Correctness-awareness may also be more important for fine-tuned models, since the casual user is less likely to interact with raw pretrained LLMs.

The second is that Kadavath et al. (2022) only study MSP calibration and not MSP correctness prediction.[4] This may be because good calibration immediately yields theoretically grounded correctness prediction, so for raw pretrained models, good calibration may be sufficient. However, our finding that chat LLMs have miscalibrated MSPs motivates the separate question of wheter MSPs can predict correctness.

The third is comprehensiveness. Kadavath et al. (2022) only test a single series of models, while we test seven series of models (14 models total). Our comprehensiveness crucially enables cross-model comparisons, as discussed in Section 1. In particular, we have statistical evidence suggesting that the correlation between correctness prediction and Q&A accuracy (and the lack of a correlation between calibration and Q&A accuracy) may extend to models that do not even exist yet. In contrast, it is harder to claim that the findings of Kadavath et al. (2022) generalize to other models, since they essentially have a sample size of one.

**Other work on LLM calibration.** Several other papers have studied MSP calibration in LLMs (OpenAI, 2023; Krause et al., 2023; Tian et al., 2023; Zhou et al., 2024). These papers have found some evidence that chat LLMs produce overconfident MSPs, but the papers use different experimental conditions and most of the papers study only a single model or family of models. To our knowledge, we are the first to present a comprehensive and unified evaluation of the MSP calibration of LLMs fine-tuned for chat.

Orthogonal to our work, Tian et al. (2023) devise prompting strategies to obtain calibrated verbalized confidence estimates from LLMs. Beyond numerical probability, recent work considers alternative media for expressing uncertainty such as natural language (Lin et al., 2022a; Mielke et al., 2022; Xiong et al., 2023; Zhou et al., 2024) or set-based similarity metrics (Lin et al., 2023).

**Training LLMs to abstain.** Another line of work fine-tunes LLMs to predict the correctness of their answers (Kadavath et al., 2022; Yin et al., 2023; Zhang et al., 2023). This approach has a different focus than our work: our goal is to understand the fundamental relationship between MSPs and correctness, not to design a state-of-the-art abstention method. Our Q&A-with-abstention experiments are intended as a simple proof of concept of our correctness prediction findings. In other words, our primary contribution is scientific, not methodological.

**Abstaining based on the MSP.** The idea of abstaining based on the MSP was originally introduced by Chow (1970) in the context of pattern recognition. This technique was recently explored for LLMs by Gupta et al. (2024), although their setting is different. Also, their experiments only use FLAN-T5 models. In contrast, we test 14 different LLMs: this enables cross-model comparisons, leading to a novel correlation between Q&A accuracy and correctness prediction AUROC.

**Beyond LLMs.** Moving further afield, uncertainty quantification is an important topic in many areas of NLP (Hu et al., 2023). Identifying data points on which a model is uncertain is a common heuristic in active learning (Settles, 2009; Tharwat & Schenck, 2023). Popular approaches rely on model output distribution, such as least confidence (Culotta & McCallum, 2005; Settles & Craven, 2008), margin sampling (Scheffer et al., 2001), and maximum entropy (Settles & Craven, 2008). Calibration (Bennet, 2000; DeGroot & Fienberg, 1983; Desai & Durrett, 2020; Guo et al., 2017; Nguyen & O'Connor, 2015; Niculescu-Mizil & Caruana, 2005) examines whether a model's predicted probability of an event accurately reflects the actual likelihood of that event occurring in

---

[4]They do study correctness prediction in the different context of training LLMs to abstain. Those results are discussed in a separate paragraph below.

the real world. In the context of error detection through model probabilities, Nguyen & O'Connor (2015) propose using MSP as an indicator for identifying incorrect predictions in coreference resolution tasks. Subsequently, Hendrycks et al. (2020) demonstrate that MSPs from pre-trained BERT models are effective in filtering out anomalous and out-of-distribution inputs. The MSP has also been used to detect mistakes and out-of-distribution inputs in vision tasks (Hendrycks & Gimpel, 2017; Hendrycks et al., 2022).

## 3 EXPERIMENT DESIGN IN FULL

As discussed, we patterned our evaluation after the original Hugging Face Open LLM leaderboard (Beeching et al., 2023). We used all five multiple-choice Q&A datasets[5] from that leaderboard: ARC-Challenge (Clark et al.), HellaSwag (Zellers et al., 2019), MMLU (Hendrycks et al., 2021), TruthfulQA (Lin et al., 2022b), and WinoGrande (Sakaguchi et al., 2021). We randomly sampled 6,000 questions from each dataset, except for those with fewer than 6,000, in which case we used all of the questions (ARC-Challenge and TruthfulQA have 2,590 and 817 questions, respectively).

We tested 14 LLMs: 12 open-source and two proprietary. The open-source LLMs were chosen based on a combination of performance on the aforementioned leaderboard and the number of downloads on Hugging Face. The open-source models we selected are Falcon (7B and 40B)[6] (Almazrouei et al., 2023), Llama 2 (7B, 13B, 70B) (Touvron et al., 2023), Llama 3 (8B, 70B) (AI@Meta, 2024), Mistral 7B v0.2 (Jiang et al., 2023), Mixtral 8x7B (Jiang et al., 2024), SOLAR 10.7B (Kim et al., 2023), and Yi (6B and 34B) (01-ai, 2023). We used the fine-tuned "chat" or "instruct" versions of all models, as these have been specifically trained to answer user queries. All of the open-source LLMs were accessed through the Hugging Face interface and were run with dynamic 4-bit quantization, which has been shown to preserve performance while significantly reducing computational requirements (Dettmers et al., 2023). The experiments on open-source LLMs took about three weeks on two NVIDIA RTX A6000 GPUs.

We also tested two proprietary LLMs for which we could obtain softmax probabilities through an API: OpenAI's GPT3.5 Turbo (Brown et al., 2020; Ouyang et al., 2022) and GPT4 Turbo (OpenAI, 2023). Unfortunately, the OpenAI API does not provide pre-softmax logits, so we could not compute Max Logit AUROCs for those two models. These experiments took only a few days and cost approximately $100.

We wanted to test our hypothesis in its simplest form, so we used a plain zero-shot prompting style, with two different prompt phrasings (Figure 1).[7] The Hugging Face Open LLM leaderboard also uses a plain prompting style (i.e., no chain-of-thought or other advanced techniques). The leaderboard does use few-shot prompting, ranging from 5-shot to 25-shot depending on the dataset. We tried using the same number of in-context examples, but this made the experiments prohibitively slow. We were able to run complete experiments with one-shot prompting, which produced similar results to our zero-shot results.

### 3.1 COMPUTING MSP AND MAX LOGIT

Let $V$ be a set of tokens (a vocabulary) and let $\boldsymbol{x}$ be a sequence of tokens from $V$. For each token $y \in V$ and prefix $\boldsymbol{x}$, an LLM computes a logit value $L(y; \boldsymbol{x})$. It then applies a softmax function to the logits to derive the probability of $y$ being the next token:

$$P(y \mid \boldsymbol{x}) = \frac{\exp(L(y; \boldsymbol{x}))}{\sum_{z \in V} \exp(L(z; \boldsymbol{x}))}$$

The Max Logit of token $y$ is $\max_{y \in V} L(y; \boldsymbol{x})$ and the MSP is $\max_{y \in V} P(y \mid \boldsymbol{x})$. Note that the Max Logit and MSP both correspond to the same token: $\arg\max_{y \in V} L(y; \boldsymbol{x}) = \arg\max_{y \in V} P(y \mid \boldsymbol{x})$.

In our experiments, we formulated each question as a prompt (Figure 1), then used greedy decoding to generate a response (i.e., we always picked the token with the maximum logit).

---

[5]The leaderboard also includes the GSM8k dataset, which we excluded since it is not multiple-choice.

[6]Falcon also has a 180B version, which we did not test due to computational constraints.

[7]Note that the two proprietary models may use advanced prompting or reasoning techniques under the hood.

For typical supervised learning classification tasks such as those studied in Hendrycks & Gimpel (2017) and Hendrycks et al. (2022), there is a single MSP and maximum logit per response. However, an LLM response can consist of multiple tokens and the model produces an MSP and Max Logit for *each token*. To address this issue, we extracted from the response a single token that indicates the LLM's answer to the question. Specifically, we searched the output string for the first occurrence of "A.", "B.", "C.", etc, then recorded the MSP and Max Logit corresponding to that capital letter token. If there was no such occurrence, we searched instead for just "A", "B", "C", etc, or the text of one of the options (e.g. for the question in Figure 1, this would be "lightning", "cellular respiration", etc.). The same search process was used for both computing MSP/Max Logit and evaluating correctness.

If the search failed, we recorded the MSP and Max Logit as zero, and labeled the LLM's response as wrong.[8] We considered excluding these questions but decided that practically speaking, these "unparseable" responses did not correctly answer the question and thus should be treated as wrong.

This search process was chosen to maximize the chance of extracting a legitimate answer from the LLM response, but the specifics of the search process (e.g., whether "A." or "lightning" has precedence) did not significantly alter the results. Most models almost always responded in the correct format. Only weaker models like Falcon 7B had issues in this regard (and even then, ambiguous answers were fairly rare).

### 3.2 AGGREGATING RESULTS ACROSS DATASETS

Grouping the questions from all datasets together to compute a single AUROC per model would undervalue datasets with fewer questions. Instead, we computed a separate AUROC for each available combination of model, dataset, prompt phrasing, and classifier (MSP vs Max Logit).[9] All in all, we recorded 260 AUROC data points (14 models $\times$ 5 datasets $\times$ 2 phrasings $\times$ 2 classifiers, excluding Max Logit for OpenAI models) and 140 Q&A accuracy data points (14 models $\times$ 5 datasets $\times$ 2 phrasings) over a total of 599,396 prompts (14 models $\times$ 21,407 questions across datasets $\times$ 2 phrasings). We then calculated per-model unweighted averages to get the results in Table 1.

## 4 PREDICTING ANSWER CORRECTNESS WITH MSP AND MAX LOGIT

Section 1.4 covers the most important elements of our AUROC results for MSP and Max Logit. We do not repeat those results here and instead proceed directly to other considerations.

**Statistical significance.** We conducted a series of hypothesis tests to confirm that our AUROC results were statistically significant. We used the Mann-Whitney $U$ test (Mann & Whitney, 1947; Wilcoxon, 1945) which directly tests the null hypothesis that the classifier's true AUROC is 50% (i.e., random guessing). In our case, a significant Mann-Whitney test affirms the hypothesis that our classifier can distinguish between (1) questions where the LLM got the correct answer and (2) questions where the LLM got the wrong answer. As the non-parametric equivalent of the $t$-test, the $U$ test requires the data to be independent but not normally distributed, which our randomly-sampled multiple choice questions satisfy.

For each available combination of model, dataset, prompt phrasing, and classifier (MSP and Max Logit), we tested the null hypothesis that the true AUROC was equal to 50%. This resulted in a total of 260 Mann-Whitney $U$ tests. Table 1 reports the number of $p$-values which were below $10^{-4}$ for each model and classifier. We chose the significance threshold of $\alpha = 10^{-4}$ to account for a Bonferroni correction (Bonferroni, 1936), whichis applied when performing multiple hypothesis tests (in our case, 220) to ensure that the chance of falsely rejecting *any* null hypothesis is small. Starting from the standard threshold of $\alpha = 0.05$, the Bonferroni correction yields $\alpha = 0.05/220 \approx 2.3 \times 10^{-4}$. We adhere to the stricter threshold of $1 \times 10^{-4}$ for simplicity.

**WinoGrande.** WinoGrande was by far the hardest dataset for our classification task (Table 2). Our best hypothesis for this discrepancy is that WinoGrande is intentionally adversarial and tries to "trick" the model. An illustrative question from this dataset is "Neil told Craig that he has to take

---

[8]This occurred for 1535 /599,396 responses (0.3%), mostly for weaker models. Across the Yi, Llama 3, and GPT models, only 50/256,884 responses (0.02%) had this issue.

[9]The Python module sklearn was used to compute AUROC.

Table 2: Average Q&A accuracy and AUROCs per dataset. All values are percentages, averaged over the then models and two prompts.

|  | Q&A Accuracy | MSP AUROC | Max Logit AUROC |
|---|---|---|---|
| ARC-Challenge | 69.5 | 71.9 | 67.2 |
| HellaSwag | 58.3 | 67.8 | 62.2 |
| MMLU | 54.4 | 68.5 | 64.1 |
| TruthfulQA | 45.8 | 66.6 | 62.0 |
| WinoGrande | 59.3 | 57.9 | 54.5 |

care of the child for the day because __ did it last time." Even for some humans, it could be unclear whether Neil is assuming responsibility or assigning responsibility. One wrinkle is that the Q&A accuracy on WinoGrande is comparable to other datasets, so it is not the case that this dataset is "harder" in general: it is harder only for predicting correctness. Regardless, despite WinoGrande's average MSP AUROC of 57.9%, GPT4 Turbo still achieved an AUROC of 77.7% on this dataset (Table 11), suggesting that this difficulty is surmountable for sufficiently capable models.

**Weak correlation between model size and AUROC.** The Pearson correlation coefficients between model size and AUROC were $r = 0.58$ ($p = 0.05$) and $r = 0.40$ ($p = 0.20$) for MSP and Max Logit, respectively (Figure 4). It is unsurprising that some correlation exists, due to the known correlation between size and model capabilities (e.g., Q&A accuracy) and our correlation between Q&A accuracy and AUROC. However, the relatively weak correlation between model size and AUROC may suggest that adding more parameters does not directly improve predictive power of these classifiers outside of improving the model's overall capabilities. The OpenAI models were excluded from these charts because we do not know their size.

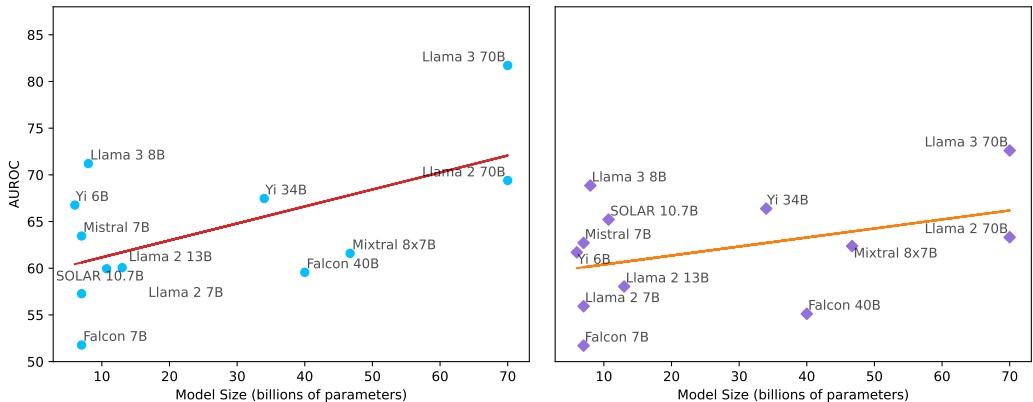

Figure 4: Average AUROC vs model size for MSP (left) and Max Logit (right). The Pearson correlation coefficients for MSP and Max Logit were $r = 0.58$ ($p = 0.05$) and $r = 0.40$ ($p = 0.20$) respectively. GPT3.5 Turbo and GPT4 Turbo were excluded since we do not know their sizes.

**Impact of prompt phrasing.** The two phrasings in Figure 1 did not yield significantly different results (Figure 5). This suggests that our results are robust to minor modifications to prompt phrasing.

## 5 PROOF-OF-CONEPT: REDUCING WRONG ANSWERS BY ABSTENTION

In Section 4, we showed that the MSP and maximum logit contain useful statistical signals for predicting correctness. To illustrate the utility of this finding, we now revisit the original Q&A task but allow models to selectively abstain based on the MSP or Max Logit.

For each classifier (MSP or Max Logit) and a given threshold, we conducted the following experiment. First, we computed the classifier value (MSP or maximum logit) based on the initial LLM

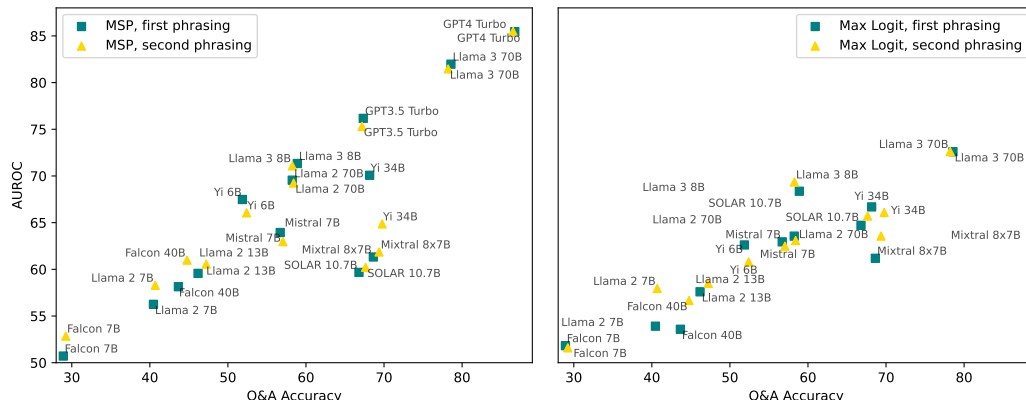

Figure 5: Average AUROC vs Q&A accuracy based on prompt phrasing (see Figure 1). All values are averaged over the five datasets.

response, the same way we did in our AUROC experiments. If the classifier value was below the threshold, we recorded the model's answer as "abstain", and otherwise recorded the original answer.

These experiments use the exact same data as the AUROC experiments, but we evaluate the results differently. We awarded 1 point per correct answer, 0 points per abstention, and $-c$ points per wrong answer, normalized by the total number of questions. We ran two versions of this experiment: once for $c = 1$ ("balanced score") and $c = 2$ ("conservative score"). For $c = 1$, the benefit of a correct answer is equal to the cost of a wrong answer. However, wrong answers are much worse in many situations (e.g., medical diagnoses), justifying $c = 2$. Our experiment design was partly inspired by Kang et al. (2024), who use an even more extreme penalty of $c = 4$.

**Choosing the threshold.** Unlike when computing the AUROC, here we must choose a specific confidence threshold for whether the model should abstain. To do so, we randomly selected a training set of $k$ questions, and then for each model, chose the threshold which performed best on those $k$ questions. We discovered that $k = 20$ performed almost as well as using half of all questions. Although $k = 10$ still outperformed the baseline, performance was much worse compared to $k = 20$. For this reason, we used $k = 20$ as the default value, but we also report results for $k = 10$ (Table 5) and $k =$ half of all questions (Table 6) in the appendix. All tables and figures other than Tables 5 and 6 use $k = 20$.

## 5.1 RESULTS

For each LLM, each classifier, and each $c \in \{1, 2\}$, Table 3 reports the scores obtained by the base LLM and the augmented LLM on the test set, where the augmented LLM used the threshold determined by training set. Figure 6 (in Appendix A) shows the performance of each model across the entire range of possible thresholds.

Our method outperformed or matched the base LLM in all conditions, and substantially outperformed the base LLM on the conservative score metric. As expected, models with low initial scores exhibited the most dramatic improvements. For example, any model with a negative initial score can trivially improve to 0 by abstaining on every question. (Table 4 shows the abstention frequency for each LLM-classifier pair.) More generally, the higher the fraction of correct answers, the more likely the model is to accidentally abstain on a correct answer. As a result, it is unsurprising that models with high initial scores showed only modest improvements on the balanced score metric.

These results show how the uncertainty signals from softmax probabilities and/or logits can be leveraged to improve performance on practical language tasks. Further details on our Q&A-with-abstention results can be found in Appendix A.

Table 3: Results on Q&A with abstention. "Balanced" and "conservative" correspond to -1 and -2 points per wrong answer, respectively. Correct answers and abstentions are always worth +1 and 0 points, respectively. The total number of points is divided by the total number of questions (then scaled up by 100 for readability) to obtain the values shown in the table. We highlight the best method for each model.

| LLM | Balanced | | | Conservative | | |
|---|---|---|---|---|---|---|
| | Base | MSP | Max Logit | Base | MSP | Max Logit |
| Falcon 7B | -41.9 | -1.1 | **-0.3** | -112.9 | -2.7 | **-1.0** |
| Falcon 40B | -11.6 | **1.9** | 0.5 | -67.4 | -0.4 | **0.0** |
| Llama 2 7B | -19.0 | -6.4 | **-5.0** | -78.5 | -6.6 | **-0.3** |
| Llama 2 13B | -6.6 | **4.9** | 4.6 | -59.9 | -2.5 | **-1.5** |
| Llama 2 70B | 16.6 | **21.4** | 19.3 | -25.1 | **7.7** | 2.3 |
| Llama 3 8B | 17.1 | **23.5** | 20.7 | -24.3 | **10.3** | 9.8 |
| Llama 3 70B | **56.8** | **56.8** | **56.8** | 35.2 | **46.5** | 42.4 |
| Mistral 7B | 13.7 | **17.0** | 15.6 | -29.4 | -1.5 | **4.8** |
| Mixtral 8x7B | 38.0 | **38.4** | **38.4** | 7.1 | **15.8** | 12.3 |
| SOLAR 10.7B | 34.4 | **35.0** | 34.4 | 1.5 | 9.4 | **13.8** |
| Yi 6B | 4.2 | **14.5** | 9.5 | -43.6 | **4.9** | -2.4 |
| Yi 34B | 37.8 | **39.6** | 38.2 | 6.8 | **18.8** | 16.4 |
| GPT3.5 Turbo | 34.5 | **38.8** | – | 1.7 | **27.0** | – |
| GPT4 Turbo | 73.3 | **73.6** | – | 60.0 | **65.2** | – |

# 6 CONCLUSION AND FUTURE WORK

The capabilities of AI systems have advanced rapidly over the past several years and will likely continue to grow. In order to ensure that AI is beneficial for society, we believe it is paramount to understand the risks of such systems and take steps to address those risks. In this paper, we focus on one particular risk: harmful responses from LLMs. We use the simple idea of noticing when one is uncertain and then acting cautiously to avert these harmful responses.

One limitation of our work is the restriction to multiple-choice questions, which simplifies the problem in several ways. First, it removes the need to distinguish between multiple correct answers ("aleatoric uncertainty") and no good answers ("epistemic uncertainty"), both of which could result in low MSPs. The restriction to multiple-choice also enabled us to tie the LLM's answer to a single token and thus a single MSP. Future work could handle free-response questions by aggregating MSPs across tokens in a clever way. A further challenge could be multi-step decision-making problems which may involve aggregating uncertainty not only across multiple tokens in a single response, but across multiple responses on different time steps.

Another limitation is our reliance on labeled data to transform our scientific insights into a practical method for abstention. We only used 20 data points, and we only used labeled data to choose the threshold, but a fully unsupervised method would be advantageous in many settings. However, we remind the reader that our primary contribution is the scientific finding of good correctness prediction despite miscalibration, not the proof-of-concept abstention experiments in Section 5.

More broadly, we are excited about developing more robust methods for mistake detection in LLMs, both for Q&A tasks and for other contexts. We would also like to better understand when and why these methods fail: are there particular subcategories of unfamiliar situations that are especially challenging to identify? For example, why was the WinoGrande dataset so much harder for our correctness prediction task?

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

## A MORE DETAILS ON Q&A-WITH-ABSTENTION EXPERIMENT RESULTS

Figure 6 is based on the same data as Table 3, but shows each model's performance across the entire range of possible thresholds. A threshold of zero corresponds to the base LLM and the black dot indicates the threshold chosen during the training phase (using 20 labeled data points), which is also the threshold used to compute the score in Table 3 using 20 data points. One can see that the chosen thresholds are not quite optimal, but 20 data points was still enough to produce substantial improvements over the baseline of not abstaining.

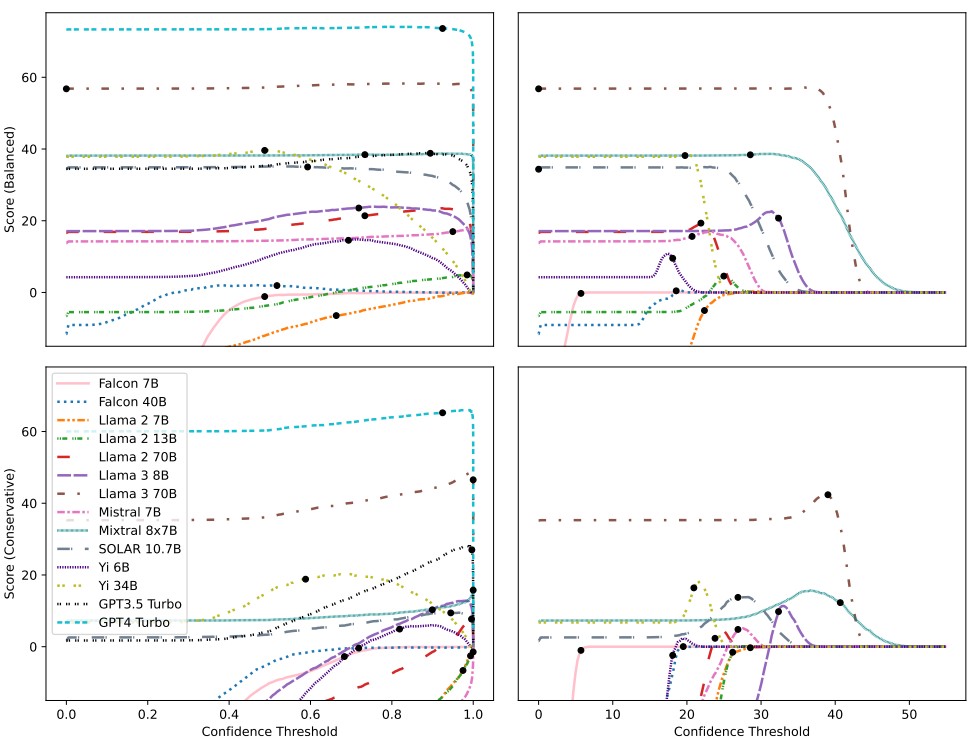

Figure 6: Results for Q&A with abstention over the entire range of possible thresholds. The base LLM corresponds to a threshold of zero. The black dot corresponds to the threshold selected via the training set, which determines the MSP and Max Logit scores in Table 3.

Table 4 presents the average abstention frequencies across all datasets, corresponding to the scores from Table 3. We also provide dataset-level versions of Table 4 in Appendix C.

Finally, Tables 5 and 6 are analogues of Table 3 using 10 data points and half of all data points for training, respectively.

Table 4: Frequency of abstention on in the Section 5 experiments.

| LLM | Balanced | | | Conservative | | |
|---|---|---|---|---|---|---|
| | Base | MSP | Max Logit | Base | MSP | Max Logit |
| Falcon 7B | 0 | 81.5 | 98.7 | 0 | 95.2 | 98.7 |
| Falcon 40B | 0 | 87.0 | 93.4 | 0 | 97.7 | 99.1 |
| Llama 2 7B | 0 | 37.2 | 47.0 | 0 | 87.3 | 98.7 |
| Llama 2 13B | 0 | 75.8 | 66.8 | 0 | 80.9 | 87.7 |
| Llama 2 70B | 0 | 14.7 | 13.8 | 0 | 66.6 | 60.8 |
| Llama 3 8B | 0 | 31.2 | 57.4 | 0 | 52.0 | 57.4 |
| Llama 3 70B | 0 | 0.0 | 0.0 | 0 | 42.4 | 25.4 |
| Mistral 7B | 0 | 15.0 | 11.7 | 0 | 49.6 | 69.9 |
| Mixtral 8x7B | 0 | 2.4 | 2.2 | 0 | 26.6 | 59.2 |
| SOLAR 10.7B | 0 | 4.7 | 0.0 | 0 | 27.5 | 35.8 |
| Yi 6B | 0 | 51.7 | 66.6 | 0 | 70.3 | 66.6 |
| Yi 34B | 0 | 10.5 | 3.9 | 0 | 25.4 | 20.6 |
| GPT3.5 Turbo | 0 | 30.8 | – | 0 | 61.2 | – |
| GPT4 Turbo | 0 | 9.6 | – | 0 | 9.6 | – |

Table 5: Q&A with abstention results using 10 questions as training data. See Table 3 for an explanation of the scoring scheme.

| LLM | Balanced | | | Conservative | | |
|---|---|---|---|---|---|---|
| | Base | MSP | Max Logit | Base | MSP | Max Logit |
| Falcon 7B | -41.9 | -1.5 | -5.2 | -112.9 | -2.7 | -1.0 |
| Falcon 40B | -11.6 | -6.2 | -6.6 | -67.4 | -3.4 | -2.6 |
| Llama 2 7B | -19.0 | -7.7 | -17.3 | -78.4 | -2.8 | -1.5 |
| Llama 2 13B | -6.6 | -1.2 | 4.6 | -60.0 | -3.6 | -7.5 |
| Llama 2 70B | 16.6 | 21.4 | 19.4 | -25.0 | 7.7 | 2.4 |
| Llama 3 8B | 17.1 | 22.6 | 17.3 | -24.3 | 10.4 | 8.1 |
| Llama 3 70B | 56.8 | 56.8 | 56.8 | 35.2 | 47.3 | 35.2 |
| Mistral 7B | 13.7 | 15.1 | 15.0 | -29.4 | -6.4 | -12.5 |
| Mixtral 8x7B | 38.0 | 38.3 | 38.2 | 7.1 | 4.9 | 12.6 |
| SOLAR 10.7B | 34.4 | 34.4 | 34.4 | 1.6 | 9.6 | 1.6 |
| Yi 6B | 4.3 | 13.2 | 10.5 | -43.6 | 5.3 | -2.4 |
| Yi 34B | 37.9 | 39.4 | 37.4 | 6.8 | 13.4 | 16.4 |
| GPT3.5 Turbo | 34.5 | 38.4 | – | 1.8 | 26.3 | – |
| GPT4 Turbo | 73.3 | 73.3 | – | 59.9 | 59.9 | – |

Table 6: Q&A with abstention results using 50% of all questions as training data. See Table 3 for an explanation of the scoring scheme.

| LLM | Balanced | | | Conservative | | |
| --- | --- | --- | --- | --- | --- | --- |
| | Base | MSP | Max Logit | Base | MSP | Max Logit |
| Falcon 7B | -41.2 | 0.0 | 0.0 | -111.8 | 0.0 | 0.0 |
| Falcon 40B | -11.1 | 1.9 | 0.7 | -66.7 | 0.0 | 0.0 |
| Llama 2 7B | -19.2 | -0.2 | 0.2 | -78.8 | 0.0 | 0.0 |
| Llama 2 13B | -6.8 | 5.1 | 4.7 | -60.3 | 1.0 | 0.1 |
| Llama 2 70B | 16.9 | 23.6 | 19.4 | -24.6 | 7.2 | 4.5 |
| Llama 3 8B | 17.2 | 23.9 | 23.1 | -24.2 | 12.6 | 11.6 |
| Llama 3 70B | 56.2 | 57.7 | 56.4 | 34.2 | 48.4 | 42.2 |
| Mistral 7B | 12.6 | 16.7 | 15.6 | -31.0 | 0.0 | 5.2 |
| Mixtral 8x7B | 37.7 | 38.6 | 38.2 | 6.6 | 15.3 | 15.0 |
| SOLAR 10.7B | 34.7 | 35.4 | 35.0 | 2.0 | 8.9 | 12.7 |
| Yi 6B | 3.1 | 13.7 | 9.9 | -45.3 | 5.3 | 2.1 |
| Yi 34B | 37.2 | 39.6 | 37.7 | 5.8 | 20.7 | 17.2 |
| GPT3.5 Turbo | 34.8 | 38.9 | – | 2.2 | 27.6 | – |
| GPT4 Turbo | 73.4 | 74.0 | – | 60.1 | 66.0 | – |

## B    CAVEAT FOR FALCON 7B

Initially, many of Falcon 7B's responses fell into the "unparseable" category described in Section 3.1. Upon investigation, we found that many of these responses were simply a period or an end-of-text token. Removing the final newline in the prompt resolved this behavior, so we believe that this newline was somehow convincing the model that was conversation was "over". These initial results had the side effect of making it very easy to detect wrong answers, since a solitary period is obviously not a correct answer. For this reason, we removed the final newline for Falcon 7B only. We considered removing the final newline for all models or excluding Falcon 7B entirely, but we felt that our chosen approach would be more scientifically honest. As Falcon 7B performed by far the worst on both Q&A and AUROC even with this concession, we do not think this decision holds much import, but we report it for transparency.

## C    DATASET-LEVEL RESULTS

For the curious reader, this section presents dataset-level versions of Table 1 (AUROC and Q&A accuracy), Table 3 (results for Q&A with abstention), and Table 4 (frequency of abstention on Q&A with abstention experiments).

Table 7: AUROC results for ARC-Challenge. See Table 7 for more explanation.

| LLM | Q&A Accuracy | MSP | | Max Logit | |
| --- | --- | --- | --- | --- | --- |
| | | AUROC | $p < 10^{-4}$ | AUROC | $p < 10^{-4}$ |
| Falcon 7B | 25.3 | 50.4 | 0/2 | 50.9 | 0/2 |
| Falcon 40B | 55.2 | 64.3 | 2/2 | 57.6 | 1/2 |
| Llama 2 7B | 45.9 | 63.4 | 2/2 | 61.3 | 2/2 |
| Llama 2 13B | 58.1 | 64.9 | 2/2 | 62.8 | 2/2 |
| Llama 2 70B | 74.4 | 77.8 | 2/2 | 69.2 | 2/2 |
| Llama 3 8B | 74.7 | 81.8 | 2/2 | 78.7 | 2/2 |
| Llama 3 70B | 92.3 | 89.7 | 2/2 | 82.3 | 2/2 |
| Mistral 7B | 70.9 | 67.9 | 2/2 | 67.8 | 2/2 |
| Mixtral 8x7B | 84.3 | 65.4 | 2/2 | 66.7 | 2/2 |
| SOLAR 10.7B | 80.8 | 61.4 | 2/2 | 70.8 | 2/2 |
| Yi 6B | 68.1 | 75.3 | 2/2 | 65.1 | 2/2 |
| Yi 34B | 84.6 | 70.7 | 2/2 | 73.2 | 2/2 |
| GPT3.5 Turbo | 83.1 | 85.1 | 2/2 | – | 2/2 |
| GPT4 Turbo | 95.7 | 88.9 | 2/2 | – | 2/2 |

Table 8: AUROC results for HellaSwag. See Table 7 for more explanation.

| LLM | Q&A Accuracy | MSP | | Max Logit | |
| --- | --- | --- | --- | --- | --- |
| | | AUROC | $p < 10^{-4}$ | AUROC | $p < 10^{-4}$ |
| Falcon 7B | 24.7 | 50.1 | 0/2 | 49.9 | 0/2 |
| Falcon 40B | 43.4 | 61.5 | 2/2 | 58.8 | 2/2 |
| Llama 2 7B | 40.9 | 58.5 | 2/2 | 54.7 | 1/2 |
| Llama 2 13B | 49.8 | 62.3 | 2/2 | 58.7 | 2/2 |
| Llama 2 70B | 64.3 | 70.0 | 2/2 | 64.4 | 2/2 |
| Llama 3 8B | 65.6 | 73.4 | 2/2 | 70.8 | 2/2 |
| Llama 3 70B | 77.7 | 81.6 | 2/2 | 70.4 | 2/2 |
| Mistral 7B | 51.8 | 62.8 | 2/2 | 62.3 | 2/2 |
| Mixtral 8x7B | 65.5 | 58.5 | 2/2 | 60.1 | 2/2 |
| SOLAR 10.7B | 78.8 | 63.1 | 2/2 | 67.1 | 2/2 |
| Yi 6B | 39.5 | 68.2 | 2/2 | 62.5 | 2/2 |
| Yi 34B | 75.5 | 73.7 | 2/2 | 67.0 | 2/2 |
| GPT3.5 Turbo | 72.5 | 76.6 | 2/2 | – | 2/2 |
| GPT4 Turbo | 88.1 | 88.3 | 2/2 | – | 2/2 |

Table 9: AUROC results for MMLU. See Table 7 for more explanation.

| LLM | Q&A Accuracy | MSP | | Max Logit | |
|---|---|---|---|---|---|
| | | AUROC | $p < 10^{-4}$ | AUROC | $p < 10^{-4}$ |
| Falcon 7B | 25.7 | 49.1 | 0/2 | 49.4 | 0/2 |
| Falcon 40B | 44.0 | 61.4 | 2/2 | 52.4 | 1/2 |
| Llama 2 7B | 40.2 | 62.2 | 2/2 | 59.3 | 2/2 |
| Llama 2 13B | 47.1 | 63.4 | 2/2 | 62.2 | 2/2 |
| Llama 2 70B | 56.3 | 71.3 | 2/2 | 64.8 | 2/2 |
| Llama 3 8B | 56.3 | 76.2 | 2/2 | 74.5 | 2/2 |
| Llama 3 70B | 75.0 | 83.5 | 2/2 | 78.6 | 2/2 |
| Mistral 7B | 52.7 | 64.5 | 2/2 | 64.3 | 2/2 |
| Mixtral 8x7B | 64.4 | 63.9 | 2/2 | 65.6 | 2/2 |
| SOLAR 10.7B | 57.4 | 62.5 | 2/2 | 68.5 | 2/2 |
| Yi 6B | 50.7 | 68.9 | 2/2 | 63.0 | 2/2 |
| Yi 34B | 64.0 | 66.0 | 2/2 | 66.2 | 2/2 |
| GPT3.5 Turbo | 64.7 | 80.5 | 2/2 | – | 2/2 |
| GPT4 Turbo | 81.4 | 85.2 | 2/2 | – | 2/2 |

Table 10: AUROC results for TruthfulQA. See Table 7 for more explanation.

| LLM | Q&A Accuracy | MSP | | Max Logit | |
|---|---|---|---|---|---|
| | | AUROC | $p < 10^{-4}$ | AUROC | $p < 10^{-4}$ |
| Falcon 7B | 19.8 | 59.5 | 1/2 | 58.1 | 0/2 |
| Falcon 40B | 26.9 | 59.3 | 1/2 | 56.5 | 0/2 |
| Llama 2 7B | 24.9 | 51.2 | 0/2 | 53.3 | 0/2 |
| Llama 2 13B | 26.4 | 57.9 | 0/2 | 54.4 | 0/2 |
| Llama 2 70B | 44.5 | 72.1 | 2/2 | 66.8 | 2/2 |
| Llama 3 8B | 40.5 | 67.8 | 2/2 | 63.8 | 2/2 |
| Llama 3 70B | 72.1 | 79.8 | 2/2 | 70.9 | 2/2 |
| Mistral 7B | 54.5 | 68.1 | 2/2 | 65.1 | 2/2 |
| Mixtral 8x7B | 67.6 | 66.4 | 2/2 | 64.9 | 2/2 |
| SOLAR 10.7B | 49.9 | 57.7 | 1/2 | 62.3 | 2/2 |
| Yi 6B | 44.2 | 64.9 | 2/2 | 62.6 | 2/2 |
| Yi 34B | 53.7 | 67.7 | 2/2 | 65.5 | 2/2 |
| GPT3.5 Turbo | 55.6 | 73.1 | 2/2 | – | 2/2 |
| GPT4 Turbo | 84.4 | 87.2 | 2/2 | – | 2/2 |

Table 11: AUROC results for WinoGrande. See Table 7 for more explanation.

| LLM | Q&A Accuracy | MSP | | Max Logit | |
|---|---|---|---|---|---|
| | | AUROC | $p < 10^{-4}$ | AUROC | $p < 10^{-4}$ |
| Falcon 7B | 49.8 | 49.8 | 0/2 | 50.2 | 0/2 |
| Falcon 40B | 51.5 | 51.2 | 0/2 | 50.4 | 0/2 |
| Llama 2 7B | 50.9 | 51.0 | 0/2 | 51.1 | 0/2 |
| Llama 2 13B | 52.1 | 51.9 | 0/2 | 52.1 | 0/2 |
| Llama 2 70B | 52.2 | 55.9 | 2/2 | 51.5 | 0/2 |
| Llama 3 8B | 55.8 | 56.9 | 2/2 | 56.5 | 2/2 |
| Llama 3 70B | 74.9 | 74.0 | 2/2 | 60.9 | 2/2 |
| Mistral 7B | 54.4 | 54.0 | 2/2 | 54.0 | 2/2 |
| Mixtral 8x7B | 63.2 | 53.7 | 2/2 | 54.6 | 2/2 |
| SOLAR 10.7B | 69.2 | 55.1 | 2/2 | 57.4 | 2/2 |
| Yi 6B | 58.1 | 56.6 | 2/2 | 55.4 | 2/2 |
| Yi 34B | 67.1 | 59.2 | 2/2 | 60.1 | 2/2 |
| GPT3.5 Turbo | 60.4 | 63.4 | 2/2 | – | 2/2 |
| GPT4 Turbo | 83.7 | 77.7 | 2/2 | – | 2/2 |

Table 12: Q&A with abstention results for ARC-Challenge. See Table 3 for an explanation of the scoring scheme.

| LLM | Balanced | | | Conservative | | |
|---|---|---|---|---|---|---|
| | Base | MSP | Max Logit | Base | MSP | Max Logit |
| Falcon 7B | -49.3 | -1.0 | 0.0 | -123.9 | -2.7 | 0.0 |
| Falcon 40B | 10.5 | 15.8 | 14.7 | -34.2 | 0.9 | 0.8 |
| Llama 2 7B | -8.1 | -2.5 | 3.8 | -62.2 | 0.9 | -16.5 |
| Llama 2 13B | 16.2 | 18.7 | 17.9 | -25.6 | 5.5 | 8.4 |
| Llama 2 70B | 48.8 | 50.3 | 50.0 | 23.3 | 39.9 | 25.2 |
| Llama 3 8B | 49.3 | 53.4 | 50.2 | 24.0 | 40.2 | 33.7 |
| Llama 3 70B | 84.6 | 84.6 | 84.6 | 76.9 | 76.9 | 76.9 |
| Mistral 7B | 41.9 | 35.2 | 37.0 | 12.8 | 22.9 | 23.2 |
| Mixtral 8x7B | 68.7 | 68.7 | 68.7 | 53.0 | 53.0 | 53.0 |
| SOLAR 10.7B | 61.5 | 61.5 | 61.5 | 42.2 | 42.2 | 42.2 |
| Yi 6B | 36.1 | 40.2 | 31.3 | 4.2 | 21.3 | 15.8 |
| Yi 34B | 69.2 | 69.2 | 65.6 | 53.8 | 53.8 | 53.7 |
| GPT3.5 Turbo | 66.3 | 67.2 | – | 49.4 | 58.1 | – |
| GPT4 Turbo | 91.4 | 91.4 | – | 87.0 | 87.0 | – |

Table 13: Frequency of abstention on ARC-Challenge in the Section 5 experiments.

| | Balanced | | | Conservative | | |
|---|---|---|---|---|---|---|
| LLM | Base | MSP | Max Logit | Base | MSP | Max Logit |
| Falcon 7B | 0 | 97.8 | 100.0 | 0 | 97.8 | 100.0 |
| Falcon 40B | 0 | 21.4 | 22.7 | 0 | 95.8 | 97.5 |
| Llama 2 7B | 0 | 14.7 | 55.7 | 0 | 87.5 | 55.7 |
| Llama 2 13B | 0 | 40.7 | 55.2 | 0 | 57.2 | 68.9 |
| Llama 2 70B | 0 | 28.9 | 7.6 | 0 | 28.9 | 55.4 |
| Llama 3 8B | 0 | 20.3 | 16.7 | 0 | 20.3 | 16.7 |
| Llama 3 70B | 0 | 0.0 | 0.0 | 0 | 0.0 | 0.0 |
| Mistral 7B | 0 | 40.2 | 33.9 | 0 | 40.2 | 46.9 |
| Mixtral 8x7B | 0 | 0.0 | 0.0 | 0 | 0.0 | 0.0 |
| SOLAR 10.7B | 0 | 0.0 | 0.0 | 0 | 0.0 | 0.0 |
| Yi 6B | 0 | 22.0 | 37.8 | 0 | 22.0 | 37.8 |
| Yi 34B | 0 | 0.0 | 14.1 | 0 | 0.0 | 18.7 |
| GPT3.5 Turbo | 0 | 2.7 | – | 0 | 26.6 | – |
| GPT4 Turbo | 0 | 0.0 | – | 0 | 0.0 | – |

Table 14: Q&A with abstention results for HellaSwag. See Table 3 for an explanation of the scoring scheme.

| | Balanced | | | Conservative | | |
|---|---|---|---|---|---|---|
| LLM | Base | MSP | Max Logit | Base | MSP | Max Logit |
| Falcon 7B | -50.5 | -30.9 | 0.0 | -125.8 | -6.3 | 0.0 |
| Falcon 40B | -13.2 | 0.9 | 0.5 | -69.7 | -1.0 | -0.3 |
| Llama 2 7B | -18.2 | -1.7 | -5.8 | -77.3 | -10.7 | -3.9 |
| Llama 2 13B | -0.3 | 6.1 | 2.4 | -50.5 | 1.2 | -1.1 |
| Llama 2 70B | 28.6 | 32.4 | 30.7 | -7.1 | 8.3 | 6.1 |
| Llama 3 8B | 31.3 | 35.4 | 32.1 | -3.1 | 18.4 | 17.1 |
| Llama 3 70B | 55.3 | 55.3 | 54.7 | 32.9 | 38.2 | 38.2 |
| Mistral 7B | 3.7 | 10.3 | 9.8 | -44.5 | -9.1 | 1.7 |
| Mixtral 8x7B | 31.1 | 29.0 | 26.7 | -3.3 | 1.5 | 3.5 |
| SOLAR 10.7B | 57.7 | 58.3 | 57.7 | 36.5 | 41.1 | 36.5 |
| Yi 6B | -20.9 | 5.7 | 0.9 | -81.4 | 1.4 | -0.1 |
| Yi 34B | 51.0 | 46.6 | 48.7 | 26.4 | 36.2 | 32.2 |
| GPT3.5 Turbo | 45.0 | 45.9 | – | 17.5 | 31.6 | – |
| GPT4 Turbo | 76.1 | 76.4 | – | 64.2 | 69.5 | – |

Table 15: Frequency of abstention on HellaSwag in the Section 5 experiments.

| LLM | Balanced | | | Conservative | | |
|---|---|---|---|---|---|---|
| | Base | MSP | Max Logit | Base | MSP | Max Logit |
| Falcon 7B | 0 | 38.6 | 100.0 | 0 | 95.0 | 100.0 |
| Falcon 40B | 0 | 95.3 | 77.1 | 0 | 95.3 | 97.1 |
| Llama 2 7B | 0 | 65.5 | 53.7 | 0 | 78.1 | 95.0 |
| Llama 2 13B | 0 | 84.2 | 20.9 | 0 | 84.2 | 85.4 |
| Llama 2 70B | 0 | 19.3 | 8.8 | 0 | 19.3 | 21.8 |
| Llama 3 8B | 0 | 30.6 | 7.0 | 0 | 30.6 | 54.4 |
| Llama 3 70B | 0 | 0.0 | 12.4 | 0 | 6.8 | 12.4 |
| Mistral 7B | 0 | 50.9 | 63.3 | 0 | 50.9 | 85.7 |
| Mixtral 8x7B | 0 | 27.0 | 35.3 | 0 | 85.6 | 80.1 |
| SOLAR 10.7B | 0 | 7.3 | 0.0 | 0 | 7.3 | 0.0 |
| Yi 6B | 0 | 62.0 | 97.0 | 0 | 92.8 | 97.0 |
| Yi 34B | 0 | 32.5 | 16.8 | 0 | 32.5 | 27.9 |
| GPT3.5 Turbo | 0 | 25.4 | – | 0 | 25.4 | – |
| GPT4 Turbo | 0 | 0.5 | – | 0 | 11.8 | – |

Table 16: Q&A with abstention results for MMLU. See Table 3 for an explanation of the scoring scheme.

| LLM | Balanced | | | Conservative | | |
|---|---|---|---|---|---|---|
| | Base | MSP | Max Logit | Base | MSP | Max Logit |
| Falcon 7B | -48.6 | -15.2 | -1.4 | -122.9 | -1.4 | -0.7 |
| Falcon 40B | -12.0 | 2.1 | -12.0 | -68.0 | -19.8 | -1.7 |
| Llama 2 7B | -19.5 | -4.6 | -3.9 | -79.3 | -37.2 | -0.6 |
| Llama 2 13B | -5.9 | -1.7 | 7.0 | -58.8 | -14.1 | -7.9 |
| Llama 2 70B | 12.5 | 18.9 | 17.3 | -31.2 | 11.3 | 4.7 |
| Llama 3 8B | 12.5 | 12.8 | 12.5 | -31.3 | 16.4 | 9.2 |
| Llama 3 70B | 49.9 | 53.1 | 49.9 | 24.9 | 43.7 | 41.2 |
| Mistral 7B | 5.5 | 12.8 | 12.7 | -41.8 | -5.8 | -2.4 |
| Mixtral 8x7B | 28.8 | 29.7 | 29.5 | -6.8 | -3.8 | -4.8 |
| SOLAR 10.7B | 14.9 | 16.6 | 21.6 | -27.7 | -3.8 | 0.7 |
| Yi 6B | 1.5 | 14.8 | 9.2 | -47.8 | 5.6 | -2.9 |
| Yi 34B | 27.9 | 30.8 | 27.9 | -8.2 | 5.2 | 12.3 |
| GPT3.5 Turbo | 29.3 | 38.2 | – | -6.0 | 24.9 | – |
| GPT4 Turbo | 62.8 | 64.5 | – | 44.1 | 55.4 | – |

Table 17: Frequency of abstention on MMLU in the Section 5 experiments.

| LLM | Balanced | | | Conservative | | |
|---|---|---|---|---|---|---|
| | Base | MSP | Max Logit | Base | MSP | Max Logit |
| Falcon 7B | 0 | 69.2 | 97.1 | 0 | 98.9 | 99.5 |
| Falcon 40B | 0 | 54.0 | 0.0 | 0 | 54.0 | 95.0 |
| Llama 2 7B | 0 | 39.4 | 49.8 | 0 | 39.4 | 97.8 |
| Llama 2 13B | 0 | 12.1 | 63.3 | 0 | 54.6 | 63.3 |
| Llama 2 70B | 0 | 66.6 | 22.9 | 0 | 73.2 | 74.0 |
| Llama 3 8B | 0 | 0.9 | 0.1 | 0 | 55.9 | 50.5 |
| Llama 3 70B | 0 | 9.1 | 0.0 | 0 | 27.2 | 36.5 |
| Mistral 7B | 0 | 27.6 | 30.3 | 0 | 50.0 | 55.7 |
| Mixtral 8x7B | 0 | 3.2 | 2.0 | 0 | 3.2 | 2.0 |
| SOLAR 10.7B | 0 | 12.4 | 31.7 | 0 | 42.3 | 36.1 |
| Yi 6B | 0 | 62.9 | 42.5 | 0 | 68.8 | 67.5 |
| Yi 34B | 0 | 15.8 | 0.0 | 0 | 19.7 | 54.6 |
| GPT3.5 Turbo | 0 | 35.3 | – | 0 | 35.3 | – |
| GPT4 Turbo | 0 | 11.5 | – | 0 | 18.4 | – |

Table 18: Q&A with abstention results for TruthfulQA. See Table 3 for an explanation of the scoring scheme.

| LLM | Balanced | | | Conservative | | |
|---|---|---|---|---|---|---|
| | Base | MSP | Max Logit | Base | MSP | Max Logit |
| Falcon 7B | -60.6 | -3.6 | -1.9 | -140.9 | -10.8 | -5.0 |
| Falcon 40B | -46.3 | -5.6 | 0.0 | -119.5 | -15.7 | -1.7 |
| Llama 2 7B | -50.8 | -34.8 | -47.4 | -126.2 | -62.1 | -0.9 |
| Llama 2 13B | -47.2 | -3.3 | -6.3 | -120.8 | -13.0 | -19.2 |
| Llama 2 70B | -11.3 | 7.2 | 1.1 | -66.9 | -11.5 | -24.5 |
| Llama 3 8B | -19.2 | -4.8 | 1.4 | -78.8 | -1.0 | -9.1 |
| Llama 3 70B | 44.2 | 46.5 | 39.6 | 16.4 | 39.0 | 28.2 |
| Mistral 7B | 8.9 | 11.8 | 16.1 | -36.6 | -5.6 | -13.6 |
| Mixtral 8x7B | 35.3 | 38.2 | 34.9 | 3.0 | 8.0 | 13.9 |
| SOLAR 10.7B | -0.5 | 4.2 | 9.0 | -50.7 | -10.0 | -9.0 |
| Yi 6B | -11.6 | -3.0 | -2.3 | -67.5 | -7.9 | -3.5 |
| Yi 34B | 6.9 | 8.8 | 6.9 | -39.6 | -34.3 | -18.2 |
| GPT3.5 Turbo | 10.9 | 19.6 | – | -33.6 | 13.3 | – |
| GPT4 Turbo | 68.9 | 68.9 | – | 53.3 | 60.3 | – |

Table 19: Frequency of abstention on TruthfulQA in the Section 5 experiments.

| LLM | Balanced | | | Conservative | | |
|---|---|---|---|---|---|---|
| | Base | MSP | Max Logit | Base | MSP | Max Logit |
| Falcon 7B | 0 | 89.2 | 95.8 | 0 | 89.2 | 95.8 |
| Falcon 40B | 0 | 85.4 | 96.5 | 0 | 85.4 | 96.5 |
| Llama 2 7B | 0 | 29.1 | 3.9 | 0 | 50.6 | 99.3 |
| Llama 2 13B | 0 | 84.0 | 80.4 | 0 | 84.0 | 80.4 |
| Llama 2 70B | 0 | 36.4 | 30.0 | 0 | 45.9 | 40.3 |
| Llama 3 8B | 0 | 29.0 | 66.0 | 0 | 81.7 | 73.4 |
| Llama 3 70B | 0 | 38.5 | 37.6 | 0 | 38.5 | 37.6 |
| Mistral 7B | 0 | 6.4 | 24.4 | 0 | 38.9 | 24.4 |
| Mixtral 8x7B | 0 | 24.0 | 25.1 | 0 | 77.1 | 56.3 |
| SOLAR 10.7B | 0 | 12.9 | 41.4 | 0 | 64.7 | 57.9 |
| Yi 6B | 0 | 24.3 | 24.1 | 0 | 63.3 | 83.5 |
| Yi 34B | 0 | 5.1 | 0.0 | 0 | 5.1 | 23.9 |
| GPT3.5 Turbo | 0 | 67.9 | – | 0 | 67.9 | – |
| GPT4 Turbo | 0 | 0.0 | – | 0 | 8.1 | – |

Table 20: Q&A with abstention results for WinoGrande. See Table 3 for an explanation of the scoring scheme.

| LLM | Balanced | | | Conservative | | |
|---|---|---|---|---|---|---|
| | Base | MSP | Max Logit | Base | MSP | Max Logit |
| Falcon 7B | -0.5 | -0.5 | -0.5 | -50.8 | -10.9 | -12.0 |
| Falcon 40B | 2.9 | 2.7 | 2.9 | -45.6 | -6.2 | -5.9 |
| Llama 2 7B | 1.8 | 2.3 | 2.3 | -47.3 | -2.1 | -3.1 |
| Llama 2 13B | 4.3 | 2.1 | 2.9 | -43.6 | -9.0 | -5.6 |
| Llama 2 70B | 4.3 | 6.3 | 3.7 | -43.5 | -13.5 | -22.6 |
| Llama 3 8B | 11.6 | 11.4 | 9.4 | -32.5 | -3.5 | 1.0 |
| Llama 3 70B | 49.8 | 46.9 | 47.3 | 24.7 | 34.0 | 25.8 |
| Mistral 7B | 8.8 | 8.0 | 3.2 | -36.9 | 0.0 | 1.0 |
| Mixtral 8x7B | 26.3 | 26.3 | 26.3 | -10.5 | -10.5 | -1.5 |
| SOLAR 10.7B | 38.3 | 35.2 | 38.3 | 7.5 | 8.0 | 10.6 |
| Yi 6B | 16.2 | 16.2 | 16.2 | -25.7 | 0.9 | -0.1 |
| Yi 34B | 34.2 | 34.2 | 34.1 | 1.3 | 8.6 | 1.7 |
| GPT3.5 Turbo | 20.9 | 20.1 | – | -18.6 | 6.9 | – |
| GPT4 Turbo | 67.4 | 66.2 | – | 51.1 | 55.0 | – |

Table 21: Frequency of abstention on WinoGrande in the Section 5 experiments.

| LLM | Balanced | | | Conservative | | |
|---|---|---|---|---|---|---|
| | Base | MSP | Max Logit | Base | MSP | Max Logit |
| Falcon 7B | 0 | 0.0 | 0.0 | 0 | 78.3 | 76.8 |
| Falcon 40B | 0 | 23.3 | 0.0 | 0 | 84.5 | 86.7 |
| Llama 2 7B | 0 | 5.4 | 1.9 | 0 | 94.3 | 92.4 |
| Llama 2 13B | 0 | 75.6 | 59.3 | 0 | 75.6 | 84.0 |
| Llama 2 70B | 0 | 54.0 | 43.5 | 0 | 54.0 | 43.5 |
| Llama 3 8B | 0 | 21.5 | 45.5 | 0 | 67.7 | 93.5 |
| Llama 3 70B | 0 | 27.2 | 9.9 | 0 | 27.2 | 9.9 |
| Mistral 7B | 0 | 17.3 | 92.4 | 0 | 100.0 | 92.4 |
| Mixtral 8x7B | 0 | 0.0 | 0.0 | 0 | 0.0 | 44.4 |
| SOLAR 10.7B | 0 | 10.6 | 0.0 | 0 | 57.5 | 39.0 |
| Yi 6B | 0 | 0.0 | 0.0 | 0 | 85.9 | 99.8 |
| Yi 34B | 0 | 1.9 | 1.1 | 0 | 25.0 | 1.1 |
| GPT3.5 Turbo | 0 | 48.8 | – | 0 | 59.5 | – |
| GPT4 Turbo | 0 | 11.5 | – | 0 | 11.5 | – |