# OpenReview forum: "Probabilities of Chat LLMs Are Miscalibrated but Still Predict Correctness on Multiple-Choice Q&A"
_ICLR.cc/2025/Conference — ICLR 2025 Conference Withdrawn Submission_

### Official Review · Reviewer_C4T3 · 2024-10-31

**Soundness:** 4
**Presentation:** 4
**Contribution:** 3
**Rating:** 6
**Confidence:** 3

**Summary:**

This paper presents a comprehensive study concerning maximum softmax probabilities (MSPs) of Large Language Models (LLMs). It reveals that these MSPs are persistently miscalibrated in the context of multiple-choice question and answer scenarios.

Among other significant findings is a strong directional correlation between the accuracy of question and answer responses and the correctness prediction of MSPs.

Furthermore, a proof-of-concept experiment is carried out in an abstain setting. In this setting, a model has the option to respond "unknown" when its confidence level is insufficient. The authors demonstrate that with merely 20 labeled examples, the selected threshold based on the MSP of the initial model can result in an improvement in the performance within the abstain context.

**Strengths:**

- The paper investigated a well - motivated question: whether the Maximum Softmax Probability (MSP) of fine - tuned Large Language Models (LLMs) is calibrated?
- In contrast to previous research, this paper examined 14 LLMs, enabling its conclusions to be generalized to future LLMs more feasibly.
- The paper clearly differentiates between calibration and correct prediction, and derives interesting conclusions regarding the relationship between the two with respect to MSP.
- The scope, definition, and empirical results are presented with remarkable clarity.

**Weaknesses:**

- As the authors have noted, the study is only applicable to multiple-choice QA, which represents a more academic environment. It remains unclear to what extent the conclusions can be extended to the more general free-text responses generated by LLMs.

- For the correction prediction task, the empirical results lack comparisons with other abstain methods - those mentioned in section 2, "Training LLMs to abstain". I understand that these alternative methods require an additional finetuning step for LLMs to learn to abstain. However, it would be beneficial to see how the MSP-based thresholding method compares with those methods that demand more supervision.

**Questions:**

N/A

---

### Official Review · Reviewer_dUDt · 2024-11-01

**Soundness:** 3
**Presentation:** 1
**Contribution:** 2
**Rating:** 5
**Confidence:** 3

**Summary:**

This paper studies if LLMs are calibrated on multiple choice questions. To investigate this question, they consider the maximum softmax probability (MSP) and aim to understand the relationship between MSP and response correctness. It shows that LLM MSP are not well calibrated and that calibration does not improve as model scores improve. However, MSPs *are* predictive of correctness, and get more predictive as model scores get better. The authors also show that this information can be turned into an effective abstention method.

**Strengths:**

This paper addresses an interesting question relating to the extent to which the probability distributions of LLMs are predictive of their answers. They study this across a wide range of LLMs. They also pilot whether this information can be used to do abstention.

**Weaknesses:**

The experiments seem generally solid, for me, the main weakness of the paper is its presentation. Specifically, the structure of the paper is quite odd. The introduction is 4 pages and presents several results with mixed details on the actual methodology, most of which is explained later. Not all the results are completely understandable without those details, and some feel poorly motivated. Some (non complete) list:
- the paper heavily uses the concept of maximum softmax probability, but this is not actually explained or introduced. While it is pretty self-explanatory, it may make sense to add at least a sentence explaining what it expresses rather than assuming the reader knows.
- In 1.1 there is talk about "the QA task", but it is unclear what task this is. On the page before it talks about 5 tasks (but also without mentioning which ones)
- In 1.1 the paper jumps immediately to the conclusion that LLMs have poorly calibrated MSPs without much further explanation to what drives this conclusion. There seems to be a clear ascending line for all the LLMs, does 'poorly calibrated' qualify anything that is not on the diagonal? Some explanation is needed to support this conclusion and to contextualise how miscalibrated this actually is.
- There is no definition or explanation of how the calibration error is computed for Figure 2. In the figure caption there is talk about "expected calibration error", but it is not clear if this is the same as the actual calibration error; in any case some words should be in the text as well such that it is understandable.
- 1.3 "before detailing our results on correctness prediction" --> didn't you just report these results in the previous section (including the hypothesis stated after)?
- In 1.2, the authors mention rigorous statistical testing, but what is this testing? Computing the AUROC?
- In 1.2 there are results about correctness prediction, then in 1.3 it is written that before detailing the results (which are already mentioned in the previous subsection), the authors hypothesise something that is also already mentioned in the previous section.
- In general I am a bit confused about subsection 1.3, as it mentions several details that I think should be relevant also for the content in 1.1 (e.g. how the models were prompted and which datasets were used)
- In 1.3 the paper says that AUROC is threshold dependent (as is calibration error), but there is no mention of threshold before so it is unclear what it refers to
- Why is 1.4 called 'results' if 1.1 and 1.2 also presented results?
- What is the motivation for using AUROC for max logit? As the authors mention, it is not a probabiity and has no notion of calibration. It should be explained why to include it.
- "We analyse a variant of the original Q&A task where models can abstain" --> What is this variant? There is some explanation on how the threshold is chosen, but it is unclear what the threshold is used for (I can guess, but it is not mentioned).

Improving those things would be a start, however, I would strongly suggest the authors to adopt a more traditional paper structure that does not include a 4-page introduction where results are explained with just not enough detail. If the authors don't prefer that, at the very least they should add some more details in the introductory section such that the results are understandable, and add some words both before 1.1 and after 1.4 that explains the organisation of the paper.

Some other points:
- I believe that the choice to look at a QA task with a constrained set of accepted answers is in fact desired for the first set of experiments. However, to make claims about abstention, I think this set-up is rather weak, as it is not clear how it could generalise to a setting with multiple correct answers that may differ across questions. How would the threshold be set in that case? And how would it be set if different questions have different length answers?
- The explanation in the paragraph "Weak correlation between model size and AUROC" is a bit circular. The authors say that the weak correlation suggests that larger models only have a better AUROC because they are generally better, and there exists a correlation between AUROC and model scores, but how do they know which one causes what?
- The reported QA accuracies are all quite low. E.g the authors report that the Winogrande scores are low because it is an adversarial dataset. However, the score reported in the paper for Winogrande  for Llama 3 8B  is 55.8, while the Llama 3 paper instead reports 75.7. This is a 20 point difference.

Lastly, a small comment:
- While in theory any evaluation dataset can be turned into a QA dataset, I would hardly call HellaSwag a QA dataset. It is a commonsense NLI dataset, most frequently formulated as a cloze task. I would suggest the authors to change the phrasing around this to avoid confusion.

**Questions:**

- In 1.2 you mark that the MSP may be predictive of correctness even if it is not the probability of correctness. Doesn't this quite directly follow from the fact that all lines in Figure 2 left are trending up?
- I am curious where the statement comes from that winogrande is an intentionally adversarial task that tries to trick the model
- I am a bit puzzled about the low Winogrande scores, in the Llama 3 paper it is reported that even Gemma 7B, not exactly the largest model, has an accuracy of 75, what do you think causes the large difference? Is it because other small models or the Llama 2 models have very low scores? (Also other scores seem low wrt what has been reported in the literature btw)
- In table 3, would it be possible to add the thresholds used?

---

### Official Review · Reviewer_QAqJ · 2024-11-02

**Soundness:** 3
**Presentation:** 2
**Contribution:** 1
**Rating:** 3
**Confidence:** 4

**Summary:**

This work studies the relationship between the probabilities (or the corresponding logits) assigned by finetuned language models to the options in a multiple choice QA setup, and the correctness of the model predictions. Prior work studied this relationship to measure the calibration error of these models and generally found that finetuned models are not well calibrated, which this work also shows, but unlike prior work, this work attempts to predict the correctness of the model predictions from the model probabilities or the logits. The main findings are that classifiers trained to predict correctness from the probabilities have high accuracy, more so when the language model does well at the original QA task. Based on this finding, it is proposed that classifier can be used to have the language model abstain from answering when it is likely that the prediction is wrong.

**Strengths:**

The paper presents an interesting finding that add to the prior work on studying language model calibration. It is shown that even though finetuned language models are not well-calibrated, a practical solution exists to disallow overconfident incorrect outputs.

**Weaknesses:**

**Details of the binary classifier are missing**: The paper does not mention what kind of a model is used for the binary classification task and what its inputs are. This might be an oversight, but it is an important omission. I hope these details can be clarified during the author response (see Questions).

**There are few takeaways from the paper**: While the paper presents an interesting finding, more related questions could have been explored within the scope of this paper, towards understanding why finetuned models are uncalibrated.  Some examples:
- How do the assigned probabilities change from base to finetuned models? Including corresponding base models as well in the study would provide useful insights.
- Related to the first weakness above, what do the binary classifiers learn? Analyzing the features or weights of the classifiers could provide insights into how finetuned models can be calibrated.
- Do the binary classifiers generalize across datasets for given language model, or even across language models? Currently the work has one classifier per LM-dataset-prompt combination. Is this required? If we cannot learn more generalizable classifiers, that tells us that not only are finetuned LMs uncalibrated, but they are uncalibrated in different ways.

**Questions:**

- What are the inputs provided to the binary classifier? Section 1.3 says "we studied a binary classification task: given a multiple-choice question and the LLM’s response, predict whether the response is correct.", but since the main idea is to be able to predict the correctness given the probabilities or the logits, one would expect those to be provided as inputs to the classifier.
- Also, what kind of a model is used for the binary classification task?'
- It can be seen in Section 3.1 that the MSP values are not normalized over the multiple choice options (i.e., the probabilities assigned to the options do not sum to 1). Is this correct? If so, it might make more sense to normalize them, because unnormalized probabilities may be affected by the surface forms of the options (e.g. "A" vs "a" vs "(a)"), and it is possible that the models may seem better calibrated after normalization.
- I understand that the main focus of this work is finetuned models, but it seems natural to also include base models in the study. Why were they not included?

---

### Official Review · Reviewer_7Vfo · 2024-11-04

**Soundness:** 2
**Presentation:** 3
**Contribution:** 2
**Rating:** 3
**Confidence:** 4

**Summary:**

This paper studies the relationship between maximum softmax probabilities (MSPs) and correctness prediction in 14 large language models (LLMs) fine-tuned for chat applications. Through extensive testing on multiple-choice Q&A datasets, the researchers found that while these models' MSPs are consistently miscalibrated (showing overconfidence), they still contain useful information for predicting when answers are correct or incorrect. Notably, they discovered a strong correlation between a model's Q&A accuracy and its ability to predict correctness using MSPs, while finding no correlation between Q&A accuracy and calibration error. This suggests that as LLM capabilities improve, their ability to predict correctness will strengthen, even though calibration issues may persist. The researchers also demonstrated a practical application of their findings by showing that models could improve their performance by selectively abstaining from answering questions based on their MSP values, using only a small amount of labeled data to determine appropriate thresholds. This research provides important insights into how chat LLMs handle uncertainty and suggests ways to make these systems more reliable by enabling them to recognize when they might be wrong.

**Strengths:**

This paper investigates an important question of LLMs' calibration, especially for chat models, which has not been comprehensively studied in previous research.

**Weaknesses:**

I think the paper's experimental setting is flawed, making the conclusion and results not convincing to me.
One technical difficulty of measuring the calibration of chat models is that the answer of chat models would include rationale and other
In L327, the author said that they searched for the letter ABCD from the outputted string and collected the probability of the letter.
However, consider a simple example here.
If you ask the model the question `What is the answer to 1+1? A:3, B:2' The model answers something like `1+1=2, thus the answer is B'. Here, the probability is P(B|`1+1=2, thus the answer is'). The answer is already given in the previous rationale. Thus, the answer is definitely B, because it is a simple matching problem. This also directly leads to the over-confident issue authors witnessed in the paper.
I think even a product of each token's probability would be a better choice here.

**Questions:**

NA

---

### Note · Authors · 2024-11-14

**Comment:**

We thank all reviewers for their comments, and we will incorporate this feedback into the next version of the paper.

**Withdrawal Confirmation:**

I have read and agree with the venue's withdrawal policy on behalf of myself and my co-authors.